# The Impact of High-Temperature Stress on the Growth and Development of *Tuta absoluta* (Meyrick)

**DOI:** 10.3390/insects15060423

**Published:** 2024-06-05

**Authors:** Junhui Zhou, Wenfang Luo, Suqin Song, Zhuhong Wang, Xiafen Zhu, Shuaijun Gao, Wei He, Jianjun Xu

**Affiliations:** 1Laboratory of Integrated Pest Management on Crops in Northwestern Oasis, Ministry of Agriculture and Rural Affairs, Institute of Plant Protection, Xinjiang Academy of Agricultural Sciences, Xinjiang Key Laboratory of Agricultural Biosafety, Urumqi 830091, China; junhuiqzhou@163.com (J.Z.); lf576263465@163.com (W.L.); suqin_song@163.com (S.S.); 18339181965@163.com (X.Z.); gaoshuaijun1998@163.com (S.G.); 2College of Plant Protection, Fujian Agriculture and Forestry University, Fuzhou 350002, China; wzhuhong@126.com

**Keywords:** *Tuta absoluta*, high-temperature stress, survival rate, growth and development

## Abstract

**Simple Summary:**

*Tuta absoluta* (Meyrick) is an invasive insect pest that poses a threat to solanaceous crops. In recent years, the occurrences of extremely high temperature have been increasing due to climate change. Therefore, it is important to study the effect of high temperature on the growth and development of *T. absoluta*. In this study, when the eggs, pupae, and adults of *T. absoluta* were exposed to high-temperature stress, the pupal stage exhibited the highest sensitivity. This was demonstrated by a significant decrease in emergence rate, longevity, and fecundity. High-temperature stress during the egg stage resulted in a cascading effect, leading to a reduction in the adult longevity. This study provides a new insight for the integrated management of *T. absoluta* under high temperature.

**Abstract:**

Insect life processes and reproductive behaviors are significantly affected by extremely high temperatures. This study focused on *Tuta absoluta*, which poses a severe threat to tomato cultivars. The effects of intense heat stress on the growth, development, oviposition, and longevity of *T. absoluta* were investigated. This investigation encompassed various developmental stages, including eggs, pupae, and adults. This study revealed that egg hatching and pupa emergence rates were significantly reduced at a temperature of 44 °C maintained for 6 h. The longevity of adults that emerged after the egg and pupal stages were exposed to 44 °C for 6 h was significantly reduced compared to the control. Notably, there was no significant variation in adult fecundity after egg-stage exposure to high temperatures. However, all treatments exhibited significantly reduced fecundity compared to the control after exposure to high temperatures during the pupal stage. Adult survival rates after exposure to 40 °C and 44 °C for 3 h were 74.29% and 22.40%, respectively, dramatically less than that of the control, which was 100%. However, no significant differences were noted in terms of longevity and egg production. These results offer a better understanding of the complex interactions between extreme temperatures and the life history traits of *T. absoluta*, thereby offering valuable insights for implementing management strategies to alleviate its impact on tomato crops in response to climate change.

## 1. Introduction

The escalating temperatures caused by global climate change pose a significant threat to biodiversity [1]. The prevalence and severity of insect infestations are projected to increase with global warming, as elevated temperatures directly impact the growth, development, and reproductive capabilities of insect pests [2]. Temperature fluctuations can exert direct effects on the physiology and biochemistry of insects, while also indirectly influencing their phenology [3]. An insect’s nervous system, muscle functioning, and immune function are compromised by the extremely high temperature, potentially resulting in a state of coma and eventual mortality [4]. In general, exposure to elevated temperatures can lead to augmented insect mortality rates and population decline [5].

Extremely high temperature often induces severe adverse effects on insect growth, development, reproduction, and survival [6]. For example, it has been shown that extreme temperature is detrimental to the growth and reproduction of Vespidae wasps [7]. Furthermore, short-term exposure to high temperatures reduces the fecundity and survival of insects [8]. Moreover, extreme heat negatively impacts insect behavior and development [9]. For example, exposure to extreme temperature during development impairs mating and reproductive behaviors in adults [10]. Individual development and population dynamics of *Elatobium abietinum* are negatively affected under high-temperature conditions [11]. High-temperature stress has been reported to cause a reduction in the fecundity and longevity of *Metopolophium dirhodum* (Walker) [12,13]. However, short-term high temperatures had no detrimental effect on the survival of *Plutella xylostella*, but reduced egg hatchability [14]. Extreme temperatures ranging from 32.3 °C to 40.4 °C could induce acute lethal effects in natural populations of *Epiphyas postvittana* [15]. The impact of high temperatures (35 °C, 38 °C, and 41 °C) on *Carposina sasakii* was examined for a duration of 14 h. Although the effect on egg hatchability was minimal, it significantly influenced adult survival, longevity, and egg production [16]. The survival and reproduction of *Calliptamus italicus* significantly increased after treatment at 33 °C for 4 h, whereas high temperatures above 36 °C adversely affected their growth and reproduction [17]. High temperatures such as 38 °C for 6 h or 42 °C for 2 h could also inhibit the growth of fall armyworm *Spodoptera frugiperd*a populations [18]. *Dastarcus helophoroides* adults could not lay eggs at 42–45 °C [19].

*Tuta absoluta* (Meyrick) is an invasive insect pest that impacts solanaceous crops such as tomatoes, brinjals, and potatoes [20,21]. *T. absoluta* adults lay their eggs on leaves or young stems, and their larvae feed on leaves, young stems, and fruits [22]. This feeding behavior and the resultant damage may serve as potential pathways for pathogen invasion, thereby leading to the occurrence of secondary diseases [23]. Tomatoes could suffer massive yield loss ranging between 80 and 100% due to *T. absoluta*, which poses a serious threat to tomato production [24]. The optimal temperature range for the growth and development of *T. absoluta* is between 25 and 30 °C. [25]. In recent years, the occurrences of extremely high temperature have been increasing due to climate change [26]. Therefore, it is important to elucidate the effect of extremely high-temperature on the growth and development of *T. absoluta*, particularly considering the scarcity of such data in the existing literature. Consequently, this paper investigated the effects of different high temperatures on the egg stage, pupal stage, and adult growth and development of *T. absoluta*. The objective was to provide a theoretical foundation for forecasting and controlling *T. absoluta* populations under extremely high-temperature conditions.

## 2. Materials and Methods

### 2.1. Insect Rearing and Temperature Selection

*T. absoluta* were collected from tomatoes from Hotan City of Xinjiang Province, China. The *T. absoluta* colony was obtained from a laboratory culture continuously reared on tomato plants in artificial climate cabinets set at 25 ± 2 °C and 50% ± 10% RH under a 16:8 h (L/D) photoperiod.

Four temperature levels were used for this experiment: 32, 36, 40, and 44 °C. For each temperature, three duration treatments were adopted: 2 h, 4 h, and 6 h. Control treatments were set at 26 °C. The total number of treatments in the experiment were 13, and each treatment was replicated 5 times.

### 2.2. The Growth and Development of T. absoluta after Exposing the Eggs to High Temperatures

Eggs laid by *T. absoluta* within 12 h were collected and counted under a microscope, and 20 eggs were placed in each Petri dish (*Φ* = 9 cm) for further experiments. The hatched larvae were selected for each treatment and reared separately in Petri dishes with fresh tomato leaves every day, and the developmental time of *T. absoluta* was observed and recorded every 24 h. The males and females were paired and nourished with 10% honey to lay eggs, and the number of laid eggs and the longevity of the adults were recorded daily.

### 2.3. The Growth and Development of T. absoluta after Exposing the Pupae to High Temperatures

The larvae were reared until pupation, and 20 pupae were selected for each treatment. The males and females were paired and nourished with 10% honey to lay eggs. The number of laid eggs and the longevity of adults were recorded daily.

### 2.4. The Effect of High Temperatures on the Growth and Development of T. absoluta Adults

*T. absoluta* adults within 12 h of emergence were subjected to high-temperature treatment; 50 adults per treatment were treated at each temperature gradient (32, 36, 40, and 44 °C) for 3 h. Control treatments were set at 26 °C. Survival rates were observed and recorded after the completion of treatment. Adults were paired and nourished with 10% honey to lay eggs, and the number of eggs laid and survival duration were recorded daily.

### 2.5. Statistical Analysis

The whole process of data analysis and image drawing was completed by using R package (Version 4.3.1) [27,28]. Two-way ANOVA was conducted to measure the effect of high-temperature exposure (i.e., temperature level and duration of exposure) on egg hatching and pupal emergence rates. Other parameters in this study were analyzed by using one-way ANOVA. If the independent variables showed significant effects on the dependent variables (*p* < 0.05), Duncan’s new multiple range test (MRT) was used to find the differences between treatments. All variance analysis procedures were run by R package. Graphical outputs were prepared by using Graph Pad Prism 9.5.

## 3. Results

### 3.1. Effects of High Temperatures on Hatching Rates of T. absoluta Eggs

The hatching rate of *T. absoluta* ranged from 31.67% to 100.00%, depending on the temperature level and exposure duration. The hatching rates decreased as the duration and temperature of exposure increased (F_4,30_ = 28.92, *p* < 0.01; Figure 1). The lowest hatching rate (31.67%) was observed when the eggs were exposed to 44 °C for 6 h.

Significant differences were observed in the developmental periods of *T. absoluta* eggs, larvae, pupae, and adults as compared to the control (Table 1). The development duration of eggs ranged from 4.00 to 4.75 d; all treatments exhibited significantly shorter durations compared to the control (F_12,391_ = 4.951, *p* < 0.05). The larval developmental periods varied from 12.33 to 14.38 d. Aside from the treatments at 36 °C for 2 h, 40 °C for 4 h, and 44 °C for 4 h, the rest of the treatments all displayed prolonged larval developmental periods compared to the control (F_12,275_ = 8.452, *p* < 0.05). In the case of a 32 °C treatment for 2 h, the pupal duration was reduced to 7.97 d, significantly shorter than the control duration (9.3 d) (F_12,212_ = 1.713, *p* < 0.05). Adult longevity was 22.46 d and 20.11 d under treatments of 32 °C for 2 h and 44 °C for 6 h, respectively, both of which were significantly shorter compared to the control (F_12,85_ = 4.340, *p* < 0.05). However, the fecundity ranging from 135.11 to 218.67 exhibited no statistically significant difference compared to the control.

### 3.2. The Growth and Development of T. absoluta after Exposing the Pupae to High Temperatures

The emergence rate of *T. absoluta* pupae decreased as the temperature (F_4,45_ = 11.06, *p* < 0.01; Figure 2) and duration of treatment increased (F_2,45_ = 12.74, *p* < 0.01). A significantly lower emergence rate of 17.50% was recorded for pupae subjected to a temperature treatment of 44 °C for 6 h, which exhibited distinct differences compared to the rates observed for those treated at temperatures of 40 °C, 36 °C, and 32 °C, as well as the control. However, no significant differences were observed among the other treatments. These findings suggest that pupal stage exposed to 44 °C for more than 6 h effectively inhibits pupae emergence.

The longevity of *T. absoluta* adults was the lowest (7.89 d) when the pupal stage was exposed to 44 °C for 6 h. The longevity of adults exposed to other thermal treatments varied between 10.82 and 17.84 days. These durations were significantly shorter compared to the control group (32.67 days) (F_4,210_ = 46.512, *p* < 0.05; Figure 3).

An increase in temperature was associated with a marked decrease in the egg-laying capacity of *T. absoluta*. Following the exposure of pupae to 32 °C for more than 2 h, the number of eggs laid by the emerged adults ranged between 34.33 and 65.67 eggs per female. This was significantly less than the average of 124.33 eggs produced per female in the control group (F_12,86_ = 3.435, *p* < 0.05; Figure 4). This finding suggests that exposure to high-temperature stress during the pupal stage considerably impaired the reproductive capacity of the adults that emerged.

### 3.3. Effects of High-Temperature Stress on Survival Rate, Longevity, and Fecundity of T. absoluta Adults

As the temperature increased, the survival rate of the adult *T. absoluta* progressively decreased. There were significant differences in the survival rates of the adults after 3 h of treatment at different temperatures (F_4,24_ =154.364, *p* < 0.05; Figure 5). The survival rate after treatment at 44 °C was significantly lower than that after the other treatments, at 22.40%, followed by 74.29% at 40 °C. There were no significant differences between the treatments at 36 °C and 32 °C and the control.

The longevity for treatments at 32 °C, 36 °C, 40 °C, and 44 °C for 3 h was 33.15 d, 32.74 d, 27.95 d, and 27.05 d, respectively, with no significant differences compared to the control (34.25 d). Similarly, there were no significant differences in the number of eggs laid between the treatments at 32 °C, 36 °C, 40 °C, and 44 °C and the control.

## 4. Discussion

The exposure of insects to extreme temperatures can compromise their capacity to adapt to the biotic and abiotic conditions in their habitats [29]. Heat stress, in particular, can trigger changes in insect behavior, morphology, life history, and physiology, which diminishes their capacity to adapt to climate change and potentially leads to population decline [30]. Our study showed a significant reduction in egg hatching rates when eggs were exposed to 44 °C for more than 6 h. Therefore, exposure to high-temperature stress may result in thermal damage which further compromises survival. Zhang et al. [18] found that the hatching rate of *Spodoptera frugiperda* eggs exposed to short-term high temperatures gradually decreased with an increase in temperature and duration of exposure. This resulted in an extended developmental period for the eggs, in line with our current study. Despite this, our study also revealed a prolonged larval developmental period without any difference in adult fecundity, contradicting a similar study [16]. This result may be related to the sensitivity of different insects to high temperatures, and the differences may be observed when assessing the temperature stress response of different life stages, insect species, experimental setups, and thermal treatments [31]. This study suggests that exposure of eggs to high-temperature stress at 44 °C for 6 h significantly reduces the longevity of the adults. Other studies have reported that the carry over effects occur at a later stage of growth and development, resulting in a reduction in adult insect survival time [32,33]. The fecundity of *Coccinella septempunctata* decreased significantly when exposed to 36 °C at the egg stage [34]. Interestingly, the present results suggest that there was no significant difference in fecundity among adults exposed to high-temperature stress during the egg stage compared to the control across all treatments. This outcome could be attributed to the phenomenon of growth compensation during the later stages of development in *T. absoluta* following exposure to high-temperature stress during its egg stage.

This study demonstrates that the rate of pupal emergence gradually decreased as temperature and treatment time increased after pupal-stage exposure to high temperature. A significant reduction in emergence rate was observed after pupal-stage exposure to 44 °C for 6 h, which is consistent with a similar study [35]. The pupae of *Grapholita molesta* were exposed to high temperatures, which increased their longevity and reduced fecundity [36]. However, our study found that the exposure of the pupal stage of *T. absoluta* to high temperature caused a significant reduction in both the fecundity and longevity of the adults. This phenomenon may be attributed to alterations in the levels of development-related hormones and modifications in gene expression [37]. In the present study, we proposed that the pupal stage of *T. absoluta* represents a developmental phase highly susceptible to high-temperature conditions. The aforementioned phenomenon not only exerts an influence on the rate of emergence, but also has implications for the subsequent developmental stage in terms of adult longevity and fecundity.

It is well known that high-temperature stress affects insect survival and affects the reproductive process and fecundity of different insect species [38]. The exposure of adults to extremely high temperatures reduces sperm functioning, leading to deceased fecundity [39]. In the present study, we showed that exposure to 40 °C for more than 3 h significantly reduced the survival rate of *T. abolsulta* adults. Interestingly, *T. absoluta* longevity decreased with increasing high-temperature stress. Longevity and fecundity, however, were not significantly different. This could potentially be attributed to the tolerance of adult moths to heat stress damage. During this study, high-temperature stress was applied to *T. absoluta* in a laboratory environment to study its effects on growth and development. It is important to note that these effects could be modified by the intricate interplay of the biotic and abiotic factors in the field. Therefore, conducting similar studies under natural conditions would contribute to a more comprehensive understanding of *T. absoluta* population dynamics under high-temperature conditions.

## Figures and Tables

**Figure 1 insects-15-00423-f001:**
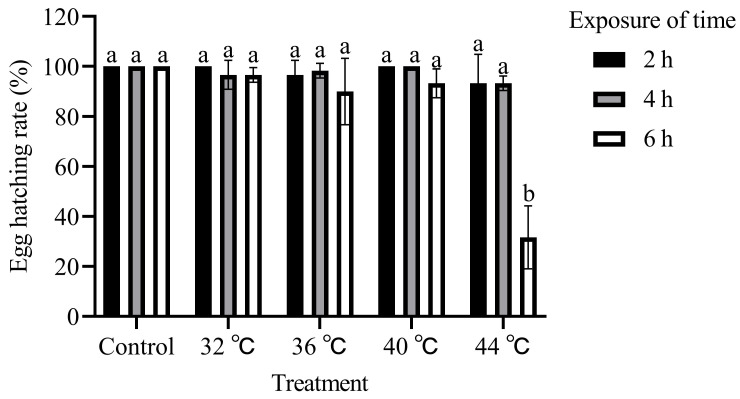
Effects of high-temperature stress on egg hatching rate of *T. absoluta*. Different lowercase letters indicate significant differences among insects subjected to various temperature treatments (*p* < 0.05, Duncan’s test). Data are expressed as means ± standard errors.

**Figure 2 insects-15-00423-f002:**
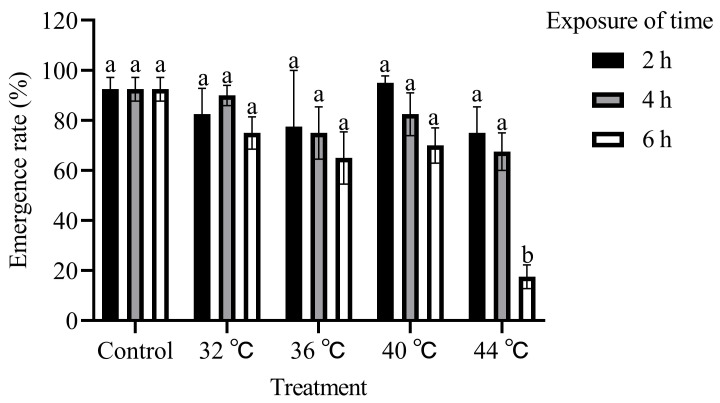
Effects of high-temperature stress during pupal stage on emergence rate of *T. absoluta*. Different lowercase letters indicate significant differences among insects subjected to various temperature treatments (*p* < 0.05, Duncan’s test). Data are expressed as means ± standard errors.

**Figure 3 insects-15-00423-f003:**
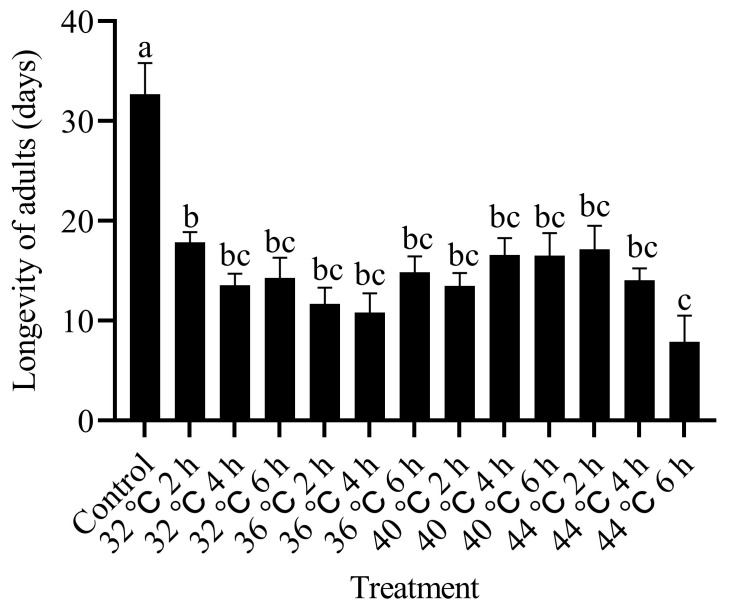
Effects of high-temperature stress during pupal stage on adult longevity of *T. absoluta*. Different lowercase letters indicate significant differences among insects subjected to various temperature treatments (*p* < 0.05, Duncan’s test). Data are expressed as means ± standard errors.

**Figure 4 insects-15-00423-f004:**
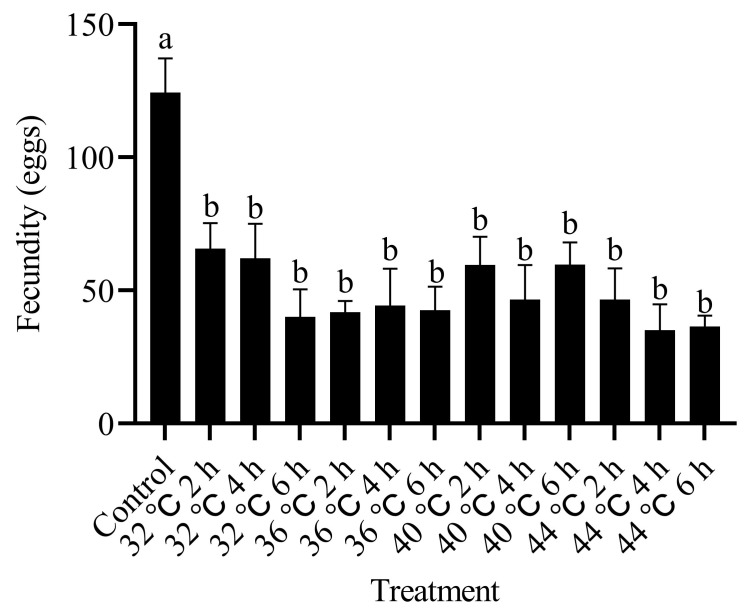
Effects of high-temperature stress during pupal stage on adult fecundity of *T. absoluta*. Different lowercase letters indicate significant differences among insects subjected to various temperature treatments (*p* < 0.05, Duncan’s test). Data are expressed as means ± standard errors.

**Figure 5 insects-15-00423-f005:**
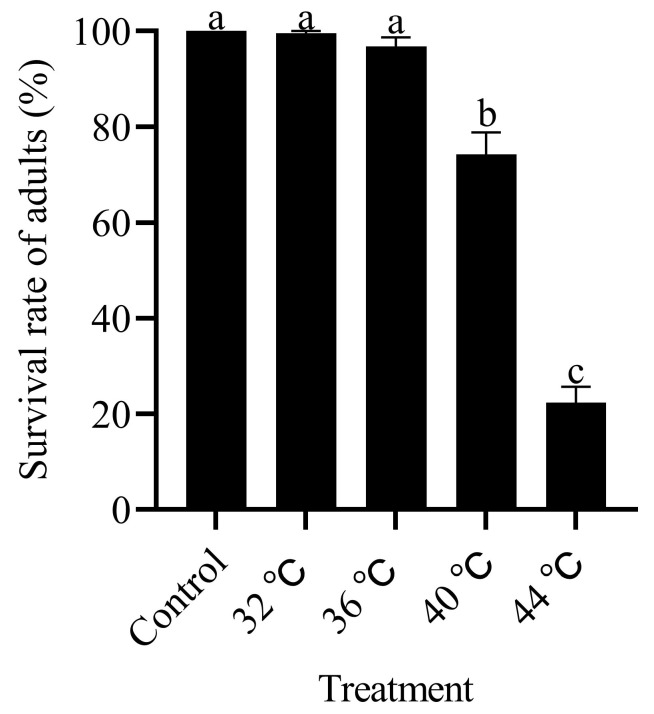
Effects of high-temperature stress on adult survival rate of *T. absoluta*. Different lowercase letters indicate significant differences among insects subjected to various temperature treatments (*p* < 0.05, Duncan’s test). Data are expressed as means ± standard errors.

**Table 1 insects-15-00423-t001:** Effects of high-temperature stress during egg stage on developmental period of *T. absoluta*.

Temperature (°C)	Time (h)	Developmental Period (Day)			Longevity (Day)
		Egg	Lavae	Pupa	Adult
32	2	4.35 ± 0.48 **b**	13.74 ± 0.81 **ab**	8.63 ± 2.03 abc	22.46 ± 7.41 **bc**
	4	4.39 ± 0.55 **b**	13.68 ± 0.95 **ab**	8.59 ± 1.00 abc	25.29 ± 8.15 abc
	6	4.32 ± 0.47 **bc**	13.33 ± 1.11 **b**	7.97 ± 1.20 **c**	25.58 ± 6.46 abc
36	2	4.22 ± 0.42 **bcde**	14.11 ± 1.83 **ab**	9.00 ± 1.33 ab	26.22 ± 8.04 abc
	4	4.43 ± 0.50 **b**	13.91 ± 1.08 **ab**	8.65 ± 1.23 abc	27.82 ± 8.95 ab
	6	4.29 ± 0.46 **bc**	13.61 ± 0.84 **ab**	8.43 ± 1.08 bc	25.53 ± 7.15 abc
40	2	4.03 ± 0.17 **de**	12.48 ± 1.58 c	9.26 ± 1.02 ab	25.46 ± 8.60 abc
	4	4.27 ± 0.45 **bcd**	12.30 ± 1.26 c	9.22 ± 0.85 ab	25.38 ± 8.48 abc
	6	4.07 ± 0.25 **cde**	13.60 ± 1.38 **ab**	9.24 ± 1.20 ab	30.25 ± 7.35 a
44	2	4.38 ± 0.49 **b**	14.38 ± 0.49 **a**	9.55 ± 1.44 a	24.45 ± 7.83 abc
	4	4.43 ± 0.50 **b**	12.33 ± 1.15 c	9.25 ± 1.14 ab	29.50 ± 6.45 a
	6	4.00 ± 0.00 **e**	13.78 ± 0.67 **ab**	9.11 ± 0.78 ab	20.11 ± 13.20 **c**
Control		4.75 ± 0.44 **a**	12.25 ± 0.72 **c**	9.30 ± 1.34 ab	30.25 ± 10.22 **a**

Data are expressed as means ± standard errors. Different lowercase letters in each column indicate significant difference in developmental periods of different stages of insects (*p* < 0.05, Duncan’s test). Bold letters indicate significant differences from the control.

## Data Availability

The data presented in this study are all available in this article.

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
