# Peer review of "The Impact of High-Temperature Stress on the Growth and Development of *Tuta absoluta* (Meyrick)"

_insects, 2024, doi:10.3390/insects15060423_

Round 1
Reviewer 1 Report
Comments and Suggestions for Authors
This report aims to investigate the effects of exposure to temperatures of 32, 36, 40, and 44 degrees Celsius for durations of 2, 4, and 6 hours on different stages of Tuta absoluta. The subsequent growth, development, longevity, and reproductive capabilities were observed and recorded post-treatment. This study provides important insights into the ecology of this pest under high-temperature conditions.
However, there is still much place for improvement in the overall presentation of this report to enhance its readability. Additionally, there are some aspects that need clarification to make the report more comprehensive.
This experiment encompassed the stages of Tuta absoluta, including eggs, pupae, and adults, all of which may experience stress from high temperatures. However, the larval stage was not included in this study, and there is no explanation provided in the research report. It is recommended to supplement an explanation for the omission of the larval stage in the study.
Especially concerning the presentation of figures and tables, there are still numerous aspects that require further modification. In most research reports, figures and tables should stand independently to clearly present experimental information, which is currently lacking in the figures and tables presented in the report.
For instance, none of the figures and tables specify whether the error bars represent standard deviation (SD) or standard error (SE). The starting value of the Y-axis scale on the figures should be 0. The lowercase English letters above the bars in the figures should denote post hoc tests, but it's unclear which treatments they are testing. This clarification should be explicitly explained in the footnote.
In LINE 99, it is not specified whether the 50 adults being treated are female or male.
The data for adults in Table 1 is no longer the developmental period but rather the longevity.
Author Response
Dear Reviewer:
Thank you for your rigorous comment. We have revised the manuscript according to your comments. Please see the attachment.
Wish you all the best!

Reviewer 2 Report
Comments and Suggestions for Authors
I found this manuscript interesting, but there are several issues with the use of English language. I had to make wide edits to improve the text and writing style. There were also some issues with the use scientific terms – I spotted many inaccurate terms throughout the manuscript. In addition, I have concerns about data analysis and data interpretation in some sections. I therefore do not recommend the publication of this manuscript in its current form. A major revision is needed before this work could be accepted for publication. Detailed comments are provided below.
Abstract
LL 12-13: “extremely high temperatures.” instead of “extreme high temperatures.”
L13: remove “the” from “the Tuta absoluta”
L14 “the effects of intense heat …” instead of “the implications of intense heat …”
LL 16-17: I do not know what the authors mean by “This study revealed a significant interaction between the temperature during treatment and the period following intense heat stress in the egg and pupal stages.” Please rephrase to improve clarity.
L 20: “was significantly reduced compared to control.” instead of “was recorded significantly less than the control.”
LL 21-22: “Notably, the number of eggs laid by adults after the egg stage was exposed to high heat showed no significant variation, but all treatments significantly exhibited fewer egg-laying events compared to control after the pupa stage was exposed to high heat.” instead of “Notably, the number of eggs laid by adults after the egg stage was exposed to high heat demonstrated no significant variation, but all treatments significantly recorded fewer egg-laying events compared to the control after the pupa stage was exposed to high heat.”
LL 25-28: “These results offer a better understanding of the complex interactions between extreme temperatures and the life history traits of T. absoluta, offering insights for management practices aimed at mitigating its impact on tomato crops in the face of climate change.” Instead of “These results enhance the understanding of the complex interactions between extreme temperatures and the life history traits of T. absoluta, offering valuable insights for pest management practices aimed at mitigating its impact on tomato crops and enhancing food security in the face of climate change.”
Introduction
L32: “The Extreme heat resulting from global climate change threatens biodiversity.” needs a supporting reference.
LL 33-34: “With global warming, the prevalence and severity of insect infestations are projected to increase, as extreme heat directly influences the growth, development, and reproduction of insect pests.” instead of “With global warming, the prevalence and severity of insect pests are projected to increase, as extreme heat directly influences their growth, development, and reproduction.”
This statement also needs a supporting reference.
LL 34-36: “Changes in temperature can directly influence the physiology and biochemistry of insects and can also impact their phenology indirectly.” instead of “Alterations in temperature can directly influence the insect physiological biochemistry, and can also indirectly impact them through changes in their phenology.”
This statement also needs a supporting reference.
L 37: musculature of muscle function?
LL 38-39: “Overall, exposure to high temperatures can cause increased insect mortality and population decline [2].” instead of “Exposure of insects to high temperatures can result in increased mortality and population decreases [2].”
LL 40-41 “Extreme heat often adversely affects the growth, development, reproduction and survival on insect species.” instead of “Extreme heat often induces severe adverse effects on insect growth, development, reproduction and survival.”
This statement also needs a supporting reference.
LL 41-42: “For example, it has been shown that extreme heat is detrimental to the growth and reproduction of Vespidae wasps [3].” instead of “Notably, it has been demonstrated that extreme heat detrimentally affects the Vespidae wasp growth and reproduction [3].”
L43: “reduces the fecundity and survival of insect species.” instead of “reduces insect fecundity and survival rates.”
L44: I am not sure what the authors mean by “significantly impacts” here, do they mean “negatively impacts”? Please clarify.
LL 45-46: how does extreme heat affect mating and reproductive behavior? Please clarify this in the text.
LL 46-47: a badly written sentence, please rewrite into a better and clearer language. Also, you need to specify how high temperature affected Elatobium abietinum. Was a negative or positive effect?
LL48-49: “High temperature stress has been reported to cause a reduction in the fecundity and longevity of Metopolophium dirhodum (Walker) [8,9].” instead of “Reduced fecundity and longevity of Metopolophium dirhodum (Walker) under high temperature stress has been reported [8,9] (Hercus et al., 2003; Ma et al., 2004).”
LL 49-51: “However, short-term high temperatures had no detrimental effect on the survival of Plutella xylostella, but reduced egg hatchability [10].” instead of LL 49-51: “However, short-term high temperatures had no significant effect on the Plutella xylostella survival, and only reduced egg hatchability [10].”
L52: “brought about acute lethal effects in the natural population …” instead of “could cause acute lethal effects in natural populations of ….”
L53: I am not sure what the authors mean by “insect stages” here. Is it different life stages? Please clarify and use accurate terminology.
L56: “significantly increased” instead of “were significantly increased”
LL 57-58: “High temperatures such as 38° C ….” instead of “High temperatures of 38° C …”
Also, please add the scientific name of grass moth to this section.
L59: “could not lay eggs at 42-45° C” instead of “could not lay eggs in 42-45° C”
The authors say that eggs were not laid at these high temperatures, then they mention that eggs did not hatch. This is contradictory logic. The authors need to rewrite this sentence to clarify the specific effects of high temperature here and remove the contraction.
L61: “an invasive insect pest” instead of “an invasive pest” and “solanaceous” instead of “lycopene”. Lycopene is an organic pigment.
LL 63-64: “This feeding behavior and the damage it brings about could provide entry points for pathogens, leading to secondary diseases.” instead of “This infests cavities and can lead to pathogen infestation resulting in secondary crop diseases.”
LL 64-66: “Tomatoes could suffer massive yield loss ranging between 80-100% due to T. absoluta, which poses a serious threat to tomato production [18].” instead of “Tomatoes could lose from 80% to 100% of their yield due to T. absoluta, posing a serious threat to tomato production [18].”
LL66-67: “The growth and development of T. absoluta is optimal when the temperature is between 25-30℃[19].” Instead of “The growth and development of T. absoluta is best suited when the temperature is 25-30℃[19].”
LL67-68: “In recent years, the occurrences of extremely high temperature has been increasing due to climate change.” instead of “Over recent years, the number of occurrences of extreme high temperature weather has been increasing annually.”
This statement also needs a supporting reference.
LL 68-71: “Therefore, it is important to elucidate the effect of extremely high-temperature on the growth and development of T. absoluta, especially that such data remain scarce in the literature” instead of Therefore, it is important to elucidate the effect of extreme high-temperature weather on the growth and development of T. absoluta. There is a scarcity of information on the growth and development of T. absoluta under 70 extreme high temperatures.”
LL 73-74: “The objective was to provide a theoretical foundation for forecasting and controlling T. absoluta populations under extremely high temperature conditions.” instead of “This aims to provide a theoretical foundation for the prediction, forecasting and green prevention and control of T. absoluta under conditions of extreme high temperature.”
Materials and Methods
L81: “climate cabinets” instead of “climatic cabinets”
LL83-86: “Four temperature levels were used for this experiment: 32, 36, 40 and 44°C. For each temperature, three duration treatments were adopted: 2 h, 4 h and 6 h. Control treatments were set at 26°C. The total number of treatments in the experiment was 13, and each treatment was replicated 5 times.” instead of “Four temperature gradients of 32, 36, 40 and 44°C were set for the short-term high temperature treatment test, and three treatment duration gradients of 2 h, 4 h and 6 h were set for each temperature, and 26°C was used as the control, with five replications of each treatment. A total of 13 treatments for this experiment.”
L87: “The growth and development of T. absoluta after exposing the eggs to high temperatures” instead of “Effect of high temperature on the egg stage on the growth and development of T. absoluta”
LL88-89: please specify the dimensions of Petri dishes.
L89: “The hatched larvae were selected…” instead of “The larvae …”
L92: “The males and females were paired and nourished …” instead of “The males and females are matched in pairs and nourished …”
L93: “the number of laid eggs and the longevity of the adults were recorded daily.” Instead of “the number of eggs laid and adults longevity are recorded daily.”
L94: “The growth and development of T. absoluta after exposing the pupae to high temperatures” instead of “Effect of high temperature during the pupal stage on the growth and development of T. absoluta”
LL 96-98: “After the adults had emerged, the males and females were paired and nourished with 10% honey to lay eggs. The number of laid eggs and the longevity of adults were recorded daily.” instead of “When the adults have fledged, the males and females are matched in pairs and nourished with 10% honey to lay eggs, and the number of eggs laid and adults longevity are recorded daily.”
L99: “Effect of high temperatures on the growth and development of T. absoluta adults” instead of “Effect of high temperature on the growth and development of T. absoluta adults”
LL 100-104: This section lack clarity. Were the adults exposed to the same temperatures and durations as in thew previous sections? Please rewrite this section to improve clarity and specify how the adults were treated.
L016: “The whole process of data analysis and images drawing was completed by R Gui (Version 4.3.1).”
Also, I am not sure what the authors mean by this. Did they use R for data analysis? If yes, they need to cite the relevant reference here. for
LL107-108: “Two-way ANOVA was conducted to measure the effect of high temperature exposure (i.e. temperature level and duration of exposure) on egg hatching and pupal emergence rates” instead of “Two-way ANOVA was conducted to measure the effect on egg hatching 107 rates and pupae emergence rates of pyroprocessing duration and temperature.”
“pyroprocessing duration” is not an accurate scientific term. It should be removed from the all the text and figures and replaced with an accurate term.
LL106-113: this section is rather vague and does lacks sufficient detail about data analysis. Was the normality of the data checked for ANOVA analysis? If yes, how? Why was Duncan’s test used? The relevant references should be cited for R packages “car” and “ggplot2”. This section should be written to address these issues.
L113: please remove “Adobe Illustrator was used in addition and picture beautification.”
Results
The number of replicates (n) and the confidence interval (CI) should be added to the stats included in the Results section. Stats should look like this: (F-value, P-value, n, CI, Figure #).
L115: “Effects of high temperatures on hatching rates of T. absoluta eggs” instead of “Effects of high temperature stress on the egg stage of T. absoluta”
LL116-117: “The hatching rate of T. absoluta ranged from 31.67% to 100.00%, depending on the temperature level and exposure duration.” instead of “The hatching rate of T. absoluta varies from 31.67% to 100.00%, contingent on teperature and stress duration.”
L120 & 121 & 122: “exposure duration” instead of “stress duration”
L122: “showed” instead of “exhibited”
LL 123-124: “The lowest hatching rate (31.67%) was observed when the eggs were exposed to 44℃ for 6 h.” instead of “The lowest hatching rate of 31.67% was observed when the eggs were subjected to high temperature stress duration under 44℃ for 6 h.”
LL 116-124: I do not see any mention of Figure 1 in this section even though Fig 1 presents the results of this section. Although the authors claim that ANOVA revealed significant differences in hatching rates among treatments, Fig 1 does not show any such significant differences. The only treatment that brought about a significant reduction in egg hatching was 44 C for 6 h. Therefore, I am concerned that the data analysis and data interpretation are erroneous.
LL127-138: An additional figure is needed to illustrate the impact of temperature on development (in addition to Table 1). Stats should also be added to this section as commented above. The relevant figure and table should be mentioned early in this section, not at the end.
L145: “The growth and development of T. absoluta after exposing the pupae to high temperatures” instead of “Effects of high temperature stress during pupal stage on the development and fecundity of T. absoluta”
LL 147-156: The authors claim that pulpal emergence rates showed significant differences among the different thermal treatments. However, Fig 3 shows no such differences among or within treatments. 44 C for 6 h should be significantly different from the rest, but the Fig 2 does not show this. Thus, I am concerned that data analysis and data interpretation for this section are erroneous.
LL159-165: “The longevity of T. absoluta adults was the lowest (7.89 d) when the pupae were exposed to 44°C for 6 h. Longevity of adults exposed to other thermal treatments varied between 10.82 - 17.84 days. These durations were significantly shorter than the control group (32.67 days) (P < 0.05; Fig. 4; all stats should be added).” Instead of The adult lifespan of T. absoluta was most curtailed, measuring just 7.89 d, when the pupal stage underwent treatment at 44°C for a period of 6 h. Lifespans for adults subject to other treatments varied between 10.82 and 17.84 days. These durations were significantly shorter compared to the control group with a lifespan of 32.67 days (P < 0.05) (Fig. 4). This suggests a trend where an increase in treatment temperature led to a progressive reduction in the adult lifespan post-exposure to high-temperature stress during the pupal stage.”
L169: “Following the exposure of pupae to 32 C for more than 2 h, …” instead of “Following a treatment duration exceeding 2h at 32 C …”
LL 171-174: all the stats should be added in this section, not only P-values.
L177: “Effects of high-temperature stress on survival rate, longevity, and fecundity of T. absoluta adults” instead of “Effects of high-temperature stress on the survival rate, lifespan and fecundity of the adult T. absoluta”
LL179-180: “As the temperature increased, the survival rate of the adult T. absoluta progressively decreased.” Instead of “As temperatures increased, the survival rate of the adult T. absoluta decreased acordingly.”
LL179-184: all the stats should be added in this section, not only P-values.
L187” “longevity” instead of “lifespan”
Discussion
L195: remove “Authors”
LL195-196: “Exposure of insects exposed to extreme temperatures can compromise their ability to adapt to biotic and abiotic the conditions in their habitats [20].” instead of “Insects exposed to extreme temperature can compromise their adaptation abilities within their respective ecosystems [20].”
LL196-198: “Heat stress, in particular, can trigger changes in insect behavior, morphology, life history, and/ physiology, which diminishes their capacity to adapt to climate change and potentially leads to population decline [21].” instead of “Heat stress, in particular, can instigate changes in their behavior, morphology, life history, and physiology diminish their capacity to adapt to climate change and potentially leading to population collapse [21].”
LL 198-200 “Our study showed that, when eggs were exposed to 44°C for more than 6 hours, egg hatching rates were significantly reduced.” instead of “Our study showed that, when eggs were exposed to 44°C for more than 6 hours significantly impacted their hatchability.”
L 201: remove “It has been shown by”
LL 209-213: I could not follow the logic here. Rephrase into better and more specific language.
L214: “when the eggs were exposed to 36 C.” instead of “when exposed to high temperatures of 36 C in the egg stage.”
L215: how insects could be feathered!! Authors should avoid blind translation and use accurate terms. Similarly, “plumage” in Line100. This is totally unprofessional!!
L 220: remove “of” from “rate of gradually”
LL 220-225 “emergence” instead of “eclosion” to keep the terminology consistent.
L 277: what are “pro-hormones”?
LL 230-231: “plumage rate” again!!
L 232: “developmental stage” instead of “metamorphosis stage”
LL 233-243: too speculative and mot supported by any empirical evidence. Better to rework this section and make it less speculative.
LL 244-258: repetitive and speculative. Better to rewrite this section, highlight the limitations of this study (it is a lab study), and speculate about what might happen in the field based on other papers that look into the impact of high temperature on the physiology and behavior of Tuta absoluta. If such papers do not exist, better to highlight this gap in knowledge and recommend the next steps for future research.
Figures
Figures are OK, but they are small. It is hard to make out any of the annotations on the figures. Figure labels are also small and need to be enlarged. The unit of the Y axis should be enclosed in round brackets.
I would prefer to see different colors in the figures (for the pyroprocessing duration), if possible. This would make the figures more crips and easier to read.
Figure 1: CK should be replaced by “Control”. The figure does not show significant different among or within treatments. The only treatment that brought about a significant reduction in egg hatching was 44 C for 6 h.
Figure 2: CK should be replaced by “Control”. The figure does not show significant different among or within treatments.
Figure 3: CK should be replaced by “Control”. The figure does not show significant different among or within treatments.
Figure 4: CK should be replaced by “Control”.
Figure 5: CK should be replaced by “Control”.
Figure 6: CK should be replaced by “Control”.
Figure 7: CK should be replaced by “Control”. The figure does not show significant different among treatments.
Tables
The table is OK, but I significant differences between treatments should be presented in bold font. Also, different temperatures should be separated by fine lines in the table.
CK should be replaced by “Control”
Use “day” instead of “d” in Development Period (d)
Comments on the Quality of English LanguageComment on language and writing style are included I the detailed comments above.
Author Response
Dear reviewer:
Thank you for your rigorous comment. We have revised the manuscript according to your comments. Please see the attachment.
Wish you all the best!

Round 2
Reviewer 1 Report
Comments and Suggestions for Authors
It still needs to be clarified that the starting value of the Y-axis scale should be 0. The authors explained, "The starting value of the Y-axis scale on all figures was 0. All aesthetic parameter settings of images we output using the ggplot2 package were already shown in the default state as well." However, in the current presentation, all the bar charts appear to be floating above the X-axis. My suggestion is that these bar charts should be directly aligned with the X-axis, which would be correct. The current presentation of the charts is quite unusual in published reports and seems incorrect. It is strongly recommended to make this correction.
Line 145 should refer to Table 1, but it currently says Table 2.
Additionally, the authors mentioned, "The data for adults in Table 1 is identified as the developmental period." It is necessary to clarify again that adults do not undergo development; this should refer to their lifespan. Therefore, the heading above the adult column should be "longevity." The current presentation of the table 1 is incorrect, and I must emphasize this again.
Author Response
Dear Reviewer:
Thanks very much for taking your time to review this manuscript. I really appreciate all your comments and suggestions. We have responded to your comments point by point.Please see the attachment.
With best regards

Reviewer 2 Report
Comments and Suggestions for Authors
I found this revised version of the manuscript improved, but there are still multiple issues with the language in addition to grammatical errors. Also, the authors did not make some of the corrections and changes I suggested in my previous review. I do not like to day this, but the authors write in a careless way sometimes, not paying attention to the quality and clarity if the language. The authors do not eve read what they write I reckon.
I cannot accept carelessly written manuscripts. Therefore, further changes and corrections are needed before the manuscript can be accepted for publication. Detailed comments are below. Once the authors make ALL the changes and corrections suggested below, and show careful and clear writing, the manuscript can be accepted.
Simple Summary
L12: “solanaceous crops” instead of “lycopene crops” lycopene is a pigment!
L14: “to study” instead of “to suggested”
LL15-16: “in this study, when the eggs, pupae, and adults of T. absoluta were exposed to high temperature stress, the pupal stage exhibited the highest sensitivity.” Instead of the current sentence.
L16: “This was demonstrated by a significant decrease in …” instead of the current.
L18: “resulted” instead of “results”
Abstract
L21: “Insect life processes …” instead of “Insects life processes …”
L25: “This study revealed that egg hatching …” instead of “This study revealed egg hatching …”
L27: “pupal stages” instead of “pupa stages”
L30: “pupal stage” instead of “pupa stage”
LL31-32: “less than the control, which was 100%” instead of “less than the control at 100%”
All T. abosulta should be inn italics.
Introduction
L43: comma before “and”
L44: “muscle functioning” instead of “musculature”
L50: remove the period after “species”
L51: “behavior” instead of “behaviors”
L52: “impairs mating and reproductive behaviors … ” instead of “reduces mating and reproductive behaviors …”
LL53-55: How the response to high temperature is affected by high temperature? Probably there is an error here. You need to fix it.
L56-57: remove “(Hercus et al., 200; Ma et al., 2004)”
L57” remove the quote mark before “However”
LL58-59: how can larvae cause acute lethal effects? The authors write in a careless way sometimes. They need to make sure that clear and accurate language is used.
L60: what is “differently insect’s states”? You need to use accurate and correct language.
L67: “at” instead of “in”
L68: “solanaceous crops” instead of “lycopene” lycopene is a pigment!!
LL69-72: you need to support these statements with references.
L107: “emergence” instead of “plumage” I commented on this in the first review, but the authors did not correct it! Plumage is for birds not insects!
L110: “after the completion of treatment” instead of the current.
LL114-121: what is “R Gui”? is it the R package? Why Duncan’s test was used?
L133: remove “CK”
LL139-140: “significantly shorter than the control (9.3d) (F…)” instead of the current.
Ll140-141: “Adult longevity was 22.46 d and 20.11 d under …” instead of the current.
L143: “between” instead of “from”
LL156-157: decreased as temperature and duration of treatment what? Increased? Also “duration of treatment” instead of “duration of treatments”
L179: “ranged between 34.33-65.67 …” instead of “from”
L182: “pupal stage” instead of “pupal phase”
Discussion
L212-213: “Exposure of insects exposed to …” “to abiotic and abiotic the conditions in their habitats” careless writing again! Please correct this.
L223: you mention “similar studies” here but you only cite one study. You need to either cite more studies here, or state that it was “a similar study” not studies.
L224: different insect species, you mean? If yes, please correct this to “insect species” instead of “insects”
L226: “life stages” instead of “age stages” what is “insect treatments”? You mean “thermal treatments”? You need to fix these errors.
L226-228: the sentence has many grammatical errors, please fix it.
L228-230: you mention “several studies” but you only cite two studies. You need to fix this by either citing more studies or using “other studies”
L231: “when exposed to 36 °C at the egg stage.” Instead of the current.
L231-L235: careless writing again with grammatical errors! Fix it.
L236: emergence rate of adults? Please clarify.
L238: what was treated at 44 °C?
L241-242: “the exposure of the pupal stage of T. absoluta to high caused a significant reduction in both fecundity and longevity of the adults. This may have been due to change in development-related hormones and changes in gene expression.” instead of the current.
L245: remove “to environmental factors and vulnerable”
L246: “impacts” instead of “impact”
L246: plumage again!!!
L249: insect species? “extremely high” instead of “extreme high”
L250: “sperm functioning” instead of “sperm function” and remove “in populations”
L251: “we showed” instead of “we indicated”
L252: the survival rate of T. abolsulta adults.” Instead of the current.
L252 “the longevity of T. abolsuta …” instead of the current.
L253: “Longevity and fecundity” instead of “Longevity and fecundity”
L255: “tolerance” instead of “robust resilience”
LL256-259: revertive – should be removed.
L259-260” remove this sentence.
L261: “to study its effects” instead of “to analyze its effects”
L262-263: “intricate interplay of the biotic and abiotic factors in the field.” Instead of the current.
L263: “Therefore” instead of “Consequently”
LL266-267: remove this sentence.
Figures
Figure 2: there are no significant difference among treatments here. Correct the figure legend or remove this figure altogether, as no significant differences were found here.
Figure 7: there are no significant difference among treatments here. Correct the figure legend or remove this figure altogether, as no significant differences were found here.
Data Availability
What do you mean by: “data sharing is not applicable”? data should be available to any researcher upon reasonable request. If not, the Editor should reject the manuscript.
Comments on the Quality of English LanguageSee detailed comments above.
Author Response

(The authors gave the same response as above.)
